# Influence of Manufacturing Defects on Mechanical Behavior of the Laser Powder Bed Fused Invar 36 Alloy: In-Situ X-ray Computed Tomography Studies

**DOI:** 10.3390/ma16082956

**Published:** 2023-04-07

**Authors:** Shuo Yang, Qidong Yang, Zhaoliang Qu, Kai Wei

**Affiliations:** 1Institute of Advanced Structure Technology, Beijing Institute of Technology, Beijing 100081, China; yang44shuo@163.com; 2State Key Laboratory of Advanced Design and Manufacturing for Vehicle Body, Hunan University, Changsha 410082, China; 2931786351@163.com

**Keywords:** laser powder bed fusion, Invar 36 alloy, manufacturing defects, mechanical behavior, in-situ X-ray computed tomography

## Abstract

The mechanical properties of laser powder bed fused (LPBFed) Invar 36 alloy have been limited by the presence of manufacturing defects. It is imperative to investigate the influence of these defects on the mechanical behavior of LPBFed Invar 36 alloy. In this study, in-situ X-ray computed tomography (XCT) tests were conducted on LPBFed Invar 36 alloy fabricated at different scanning speeds to examine the relationship between manufacturing defects and mechanical behavior. For LPBFed Invar 36 alloy fabricated at a scanning speed of 400 mm/s, the manufacturing defects were randomly distributed and tended to be elliptical in shape. Plastic deformation behavior was observed, and failure initiated from defects inside the material resulting in ductile failure. Conversely, for LPBFed Invar 36 alloy fabricated at a scanning speed of 1000 mm/s, numerous lamellar defects were observed mainly located between deposition layers, and their quantity was significantly increased. Little plastic deformation behavior was observed, and failure initiated from defects on the shallow surface of the material resulting in brittle failure. The differences in manufacturing defects and mechanical behavior are attributed to changes in input energy during the laser powder bed fusion process.

## 1. Introduction

Invar 36 alloy, which is an iron-nickel alloy with a composition in weight of 64% Fe and 36% Ni, is well known for its extremely low coefficient of thermal expansion (CTE) below Curie temperatures [1,2,3,4,5,6]. Benefiting from its extremely low expansion over a wide range of service temperatures, it has been extensively used as high precision material in engineering applications requiring high dimensional stability, such as precision instruments, space equipment, metrology devices, composite molds, and optical parts, etc. [1,2,5,6,7]. Conventionally, the Invar 36 alloy components are always manufactured by machining. However, Invar 36 alloy is a single-phase austenite with a face-centered cubic crystal structure and has high work hardening and ductility, leading to its poor machinability. Due to its high built-up edge formations, the material easily adheres to the cutting tools, which seriously affects the efficiency and precision. Thus, machining bulk Invar 36 alloy material into complex-shaped components is a particularly challenging task [8,9,10].

Fortunately, the increasingly popular additive manufacturing provides a viable solution for the efficient and highly accurate fabrication of complex-shaped components. Among various additive manufacturing technologies, laser powder bed fusion (LPBF) has shown great potential for the preparation of Invar 36 alloy [11,12]. Due to the high cooling rates, the as-LPBFed materials exhibit refined solidified microstructures and excellent mechanical properties. However, manufacturing defects including pore defects, partially melted or unmelted powder and macro/micro-cracks, etc are inevitably induced during the LPBF process. The presence of manufacturing defects is detrimental to the mechanical properties and significantly affects the failure behavior [13,14]. Qiu et al. [12] studied the influence of pore defects on the mechanical properties of Invar 36 alloys fabricated at the vertical and horizontal directions. Pore defects had different effects on the elongation, ductility, and tensile strength of vertically and horizontally fabricated specimens due to the tensile/opening loading mode incurred. Yakout et al. [15,16] pointed out that the formation of internal pores when melted at laser energy density lower than brittle-ductile transition energy density resulted in the brittle fracture. Wegener et al. [7] suggested that defects in the LPBFed specimen with a relative density of 99.6% introduced a negligible influence on the low cycle fatigue life. The above studies mainly focused on the characterization of defects, microstructures, mechanical tests, and fractographic analysis. Indeed, the characterization of the damage evolution during mechanical loading is also crucial for the analysis of mechanical behavior. Unfortunately, few studies have focused on the defect evolution of LPBFed Invar 36 alloy during mechanical loading. Further, the information on the critical defects that lead to the ultimate failure is rarely obtained. The influence of manufacturing defects on the mechanical behavior of LPBFed Invar 36 alloy remains to be further investigated.

X-ray computed tomography (XCT) technology, which can obtain three-dimensional full-field information on materials [17,18,19,20,21], has been used to characterize manufacturing defects in the as-LPBFed Invar 36 alloy [22,23]. In order to characterize the damage evolution during mechanical loading, in-situ XCT technology was developed by integrating the conventional XCT and the mechanical loading, widely adopted in the research of composites [24,25], ceramics [26,27], superalloys [28,29], etc. However, the study on the characterization of LPBFed Invar 36 alloy by using in-situ XCT technology are still not conducted.

In this study, in-situ XCT tests were conducted on the LPBFed Invar 36 alloy fabricated at different scanning speeds, and the evolution behavior of manufacturing defects under tensile loads was obtained. The critical defects that induce failure were located, and digital volume correlation (DVC) were also adopted to further study the influence of manufacturing defects on mechanical behavior. The influence mechanisms of the materials at different scanning speeds were discussed in detail.

## 2. Experimental and Methods

### 2.1. Materials and Specimens

In this study, Renishaw AM250 (Renishaw, Gloucestershire, UK) with an ytterbium fiber laser was used to prepare Invar 36 alloy. The gas-atomized Invar 36 virgin powder with the diameter of the powder ranges from 20 to 50 μm was provided by AMC powders Corporation (Beijing, China). The process parameters involved in the LPBF process include the laser beam diameter d, laser power P, hatch spacing h, layer thickness t, and average scanning speed v. The laser beam diameter was set to 70 μm, the laser power to 200 W, the hatch spacing to 90 μm, and the layer thickness to 30 μm. To investigate the influence of scanning speed on the defect evolution behavior, the scanning speed was designed as 400 and 1000 mm/s, and the linear energy densit (LED) corresponding to the scanning speed is *E_l_* = 0.5 J/mm and *E_l_* = 0.2 J/mm (*E_l_* = *P*/*v*), respectively. As illustrated in Figure 1, the Invar 36 alloy plates were fabricated by LPBF, and then the plates were cut into specimens by using the electric discharge machining (EDM) wire-cutting method. The specific size of the sample is shown at the bottom of Figure 1. In order to distinguish the specimens under different parameters, the specimens fabricated at the scanning speed of 400 and 1000 mm/s were named Specimen 400 and Specimen 1000, respectively.

### 2.2. In-Situ Tensile Tests

In-situ tensile tests were conducted on Invar 36 alloy specimens using an in-situ X-ray computed tomography apparatus assembled in our laboratory [30]. This apparatus consists of a microfocus X-ray source that produces X-rays, a CCD detector that receives X-rays, as well as a material testing machine with two synchronous rotating motors for mechanical loading and 360-degree rotation of the specimen. The tensile test was divided into two stages. Firstly, the specimens were loaded to 50 N for the initial (unloaded) scan. Subsequently, tensile loads were applied to the specimens using a displacement control mode, and the tensile tests were interrupted in three times to conduct tomographic scans. The three interruptions are the point occurs yield, the point have occurred large deformation, and the point near to break, respectively. All of these points were obtained from static tensile experiment. For each scan process, the tube voltage and current was 200 kV and 100 μA, respectively. The exposure time was 3 s per projection, and 1000 projections were recorded. The total scan time was 50 min, and the effective voxel size was 5 μm. To obtain the fracture morphology, the tomographic scans on the region near the fracture were conducted after the specimen failed. Reconstructed images were obtained using a filtered back-projection algorithm, and the image processing software Avizo was used for three-dimensional visualization, segmentation, and quantification. Afterwards, scanning electron microscope (SEM) tests were performed on the fractured specimen with a Quanta 250 FEG (Hillsboro, OR, USA) microscope to investigate more detailed fracture morphology. The Quanta 250 FEG microscope is field emission electron microscope and the type of detector is secondary.

### 2.3. Digital Volume Correlation (DVC) Analysis

To further study the mechanical behavior of LPBFed Invar 36 alloy, digital volume correlation (DVC) method was used to obtain the displacement and strain field based on the reconstructed CT images. The DVC analysis was performed using the Digital Volume Correlation module embedded in the AVIZO 2020.1 software. To reduce the computational workload, the volume near the fracture was selected as Volume of Interest1 (VOI1), with a size of 669 voxels × 542 voxels × 732 voxels. To improve the efficiency and accuracy of the computations, the local DVC method was used to obtain the rough three-dimensional deformation field of the sub-volume with a size of 400 voxels × 400 voxels × 400 voxels. Subsequently, the displacement field obtained by the local DVC method was used as the initial value for the global DVC method. A tetrahedral mesh with a size of 80 voxels was used in the global DVC method. The maximum number of iterations was 30 and the convergence criterion was 0.001 voxel.

## 3. Results

### 3.1. Load-Displacement Curves

The load-displacement curves at different scanning speeds are plotted in Figure 2. It can be seen from Figure 2 that the Invar 36 alloy exhibits typical plasticity. The yield loads of Specimen 400 and Specimen 1000 are 4138 and 3472 N, respectively, and their peak load are 4810 and 3802 N, respectively. The fracture displacements of Specimen 400 and Specimen 1000 are 2.83 and 0.98 mm, respectively. It was found that the high scanning speed results in a reduction on the yield strength and an improvement on the fracture displacement suggesting that the plasticity of Specimen 400 is more pronounced than that of Specimen 1000. The ultimate strength, on the other hand, is negatively related to the scanning speeds, indicating that the load capacity of Specimen 400 is greater than that of Specimen 1000. The solid circles in Figure 2 indicate the scan points at different loading steps, which were named sequentially as step0, step1, step2, and step3 with increasing tensile displacements. The relaxation phenomenon was observed during the constant displacement (holding) periods at each scan point.

### 3.2. Three-Dimensional Volume Rendering

The three-dimensional volume rendering was obtained by stacking the two-dimensional CT slices. The volume near the fracture was selected as the Volume of Interest2 (VOI2), which had dimensions of 669 voxels × 542 voxels × 664 voxels. There are some differences between VOI1 and VOI2. VOI1 is used for DVC computation, and it contains a part of the hard-to-deform area to ensure accurate measurement of the amount of deformation. VOI2 is used to extract defects in the area prone to deformation and does not contain the hard-to-deform area. Volume renderings of VOI2 at different loading steps are shown in Figure 3, the defects inside the specimens are highlighted in blue. For Specimen 400, few defects with a random distribution are observed, while a large number of defects accumulate among deposition layers in Specimen 1000. This phenomenon is consistent with the result in the reported literature [12]. The profiles of VOI2 at different loading steps were also studied. It is found that the length of VOI2 along the loading direction increases significantly with increasing tensile displacements, and obvious necking is observed for Specimen 400. For Specimen 1000, a much smaller variation in the length with no significant necking of VOI2 can be observed, indicating that Specimen 400 tends to exhibit plasticity more than Specimen 1000.

### 3.3. Defect Evolution Behavior

To quantitatively study the defect evolution behavior under tensile loads, the porosities were measured at different loading steps, as shown in Figure 4. It can be found that the porosity of Specimen 1000 is much larger than that of Specimen 400, as some of the powder failing to melted sufficiently during specimen preparation at 1000 mm/s. The porosities of both Specimen 400 and Specimen 1000 increased with increasing tensile displacements, indicating that pore expansion occurs in specimens during loading.

To thoroughly investigate the influence of manufacturing defects on mechanical behavior, the study focused on the evolution behavior of critical defects that lead to ultimate failure. Figure 5 shows the complete evolution process of critical defects under tensile loads, and CT slices containing critical defects were extracted and are highlighted by rectangles. For Specimen 400, the critical defect is located inside the specimen and has a small volume. At step0, the shape of the critical defect is nearly elliptical. The volume of the critical defect was slightly changed from step0 to step1 due to the elastic deformation. The volume was sharply increased from step1 to step2 owing to the large plastic deformation, and continued to increase from step2 to step3. For Specimen 1000, the critical defect is located on the surface of the specimen and manifests as a lamellar defect among the deposition layers and perpendicular to the loading direction. The volume of the critical defect increases with increasing tensile displacements, and a significant increase in size along the loading direction is observed.

### 3.4. Fractography

After the specimen failed, the area near the fracture was scanned, and the resulting volume renderings are presented in Figure 6. For Specimen 400, the cross-sectional area of the fracture is smaller than that of other parts, which is a result of the necking behavior. The fracture is not flat, indicating a ductile fracture mode. In contrast, for Specimen 1000, the cross-sectional area of the fracture is similar to that of other parts, and the fracture is relatively flat, indicating a brittle fracture mode.

## 4. Discussion

### 4.1. Deformation Mechanism

According to Figure 3, the lengths of VOI2 along the loading direction at different loading steps were calculated, and the elongation ratios at different loading steps were obtained. Additionally, the cross-sectional areas of the specimen perpendicular to the loading direction at different loading steps were calculated, and the reductions of area for VOI2 at different loading steps were obtained and shown in Figure 3b. It was found that the reductions of area for Specimen 400 and Specimen 1000 both decrease with increasing tensile displacements. For Specimen 400, the elongation ratio and reduction in area change little from step0 to step1, indicating that elastic deformation occurs in this stage. The elongation ratio and reduction in area sharply increase from step1 to step2, due to plastic deformation dominating in this stage. The elongation ratio and reduction in the area continue to increase from step2 to step3. For Specimen 1000, the elongation ratio and reduction in the area both increase linearly, indicating that the deformation of the specimen is mainly elastic, and little plastic deformation occurs before fracture. As an austenitic alloy, Invar alloy has high ductility [15]. Figure 7 indicates that high scanning speed will damage this property, and v = 400 mm/s is a scanning speed to maintain this property.

### 4.2. Defect Evolution Mechanism

As mentioned earlier, the characteristics of the defects inside Specimen 400 are different from those of the defects inside Specimen 1000. It is well known that sphericity is a common parameter to evaluate the shape of an object. The closer the sphericity is to 1, the closer the shape of an object is to a sphere. Thus, sphericity can be adopted to characterize the shape of the defects. In general, the sharp shape is prone to induce stress concentration. The sphericity *S* can be determined as follows:(1)$$S=\frac{{{\left(3V_{p}\right)}^{\frac{2}{3}}\cdot (4\pi )}^{\frac{1}{3}}}{S_{p}}$$

The volume $V_{P}$ and surface area $S_{P}$ of each defect were obtained from AVIZO software, and the equivalent diameter d of each defect was calculated by equating it to a sphere with the same volume.

The relationships between the equivalent diameter *d* and sphericity *S* for Specimen 400 and Specimen 1000 at step0 and step3 were extracted and are shown in Figure 8. The number of defects especially large volume defects (*d* > 60 μm) in Specimen 1000 is higher than that in Specimen 400. At the lower scanning speed (*v* = 400 mm/s), the temperature field reaches a higher peak value, leading to a larger size of the molten pool. Therefore, Specimen 400 mainly contains metallurgical pores and a few keyhole pores with a diameter greater than 60 μm. As shown in Figure 8a, the defect shape in this specimen is more regular in terms of sphericity. As the scanning speed is up to be v = 1000 mm/s, since the input energy density decreases, the peak temperature field decreases and the lap area being insufficient, resulting in more powder cannot be melted. The defects in specimen 1000 are mainly lack of fusion (LOF) pores. The sphericity of some large defects (*d* > 100 μm) is small (*S* < 0.5) as shown in Figure 8c, suggesting that LOF pores are more complex in shape, and their edges are sharper.

Most of the defects in both Specimen 400 and Specimen 1000 at step0 have small equivalent diameters. Additionally, defects with sphericity close to 1 typically have small volumes at step0, while those with large volumes typically have low sphericity, which increases the possibility of specimen failure during loading. The volume of defects increases with increasing tensile displacements, especially for the largest volume defect. For Specimen 400, only a small number of large volume defects (*d* > 100 μm) appear at step3. Most defects show an increase in sphericity with increasing load, with a decrease in the number of defects with small sphericity (*S* < 0.6). After loading, a defect with a significantly larger volume than others is generated in the specimen with a sphericity of 0.8. In contrast, for Specimen 1000, the number of large volume defects increases significantly at step3, and their sphericity decreases further. The phenomenon indicates that the pore change in specimen 400 is the expansion of a single defect, while defect penetration occurs in specimen 1000.

To provide more insight into the evolution of defect morphology, three-dimensional morphology of the critical defects was extracted and plotted in Figure 9. For Specimen 400, the volume of the critical defect is small and the shape of the defect tends to be an ellipsoid with the long axis perpendicular to the loading direction at step0. The volume and shape of the critical defect are changed slightly from step0 to step1, owing to that elastic deformation occurs in this stage. The volume of the critical defect sharply increases from step1 to step2, and expands rapidly in all directions. The long axis of the ellipse to be parallel to the loading direction, resulting from that large plastic deformation occurs in this stage. In addition, the volume of the critical defect continues to increase from step2 to step3. For Specimen 1000, the critical defect is manifested as lamellar defects, which are mainly located among the deposition layers and perpendicular to the loading direction. When the laser scanning speed V = 400 mm/s and 1000 mm/s, the depth of molten pool is 180 μm and 50 μm, respectively. Both of them are greater than the powder layer thickness (30 μm). However, considering that the shape of the molten pool is arc-shaped, the bottom of the molten pool may be insufficiently overlapped under the v = 1000 mm/s process, resulting in insufficient powder melting. The volume of the critical defect increases with increasing tensile displacements. The defect extends in the plane perpendicular to the loading direction and merges with small defects nearby. A significant increase in size along the loading direction is also observed.

### 4.3. Failure Mechanism

The three-dimensional deformation field was obtained to investigate the failure behavior of the specimen. The CT data at step2 and step3 were used for DVC calculation, and the displacement (W) and uniaxial strain (zz) field of the VOI1 were calculated and shown in Figure 10. For Specimen 400, the uniaxial strain on the surface of the specimen is relatively uniform, and the stress concentration region is located above the center of the specimen. Thus, severe pore tearing occurred at this location as shown in Figure 5a. The 3D rendering image of fracture, as shown in Figure 10c, shows an obvious dimple, which is similar to the pore in Figure 5a. It can be considered that such manufacturing defects aggravate the deformation behavior. For Specimen 1000, there is a severe strain concentration on the upper left of the center of the specimen. The strain concentration region is located near the fracture of the specimen, indicating that the fracture is mainly caused by the local strain concentration.

The fracture of Specimen 400 varied from Specimen 1000. In detail, a few small spherical pores are sparsely distributed inside Specimen 400, as shown in Figure 3a. The inclined shear surface coupled with the obvious necking phenomenon present at the fracture, as shown in Figure 6a,c. Ductile dimple, symbolizing ductile failure, can be found in Figure 11b. Minor amounts of scattered pores indicates that pore aggregation does not occur. For Specimen 1000, a large amount of LOF defects appear in the flat fracture surface perpendicular to the loading direction. As shown in Figure 6b,d, there is almost no necking deformation in the fracture, indicating the brittle failure mode. A large amount of unmelted powder can be found in Figure 11d. The defects can be observed on the fracture surface, and the shape of the defect is irregular lamellar depression. It indicates that insufficient energy input caused the powder to fail to be melted and eventually generated numerous LOF defects. The aggregation of LOF defects leads to the propagation of cracks between layers.

## 5. Conclusions

Herein, the influence of manufacturing defects on mechanical behavior of LPBFed Invar 36 alloy was studied by in-situ XCT. The specimens were fabricated at two different laser scanning speeds, there are 400 and 1000 mm/s. For Specimen 400, the defects observed are small in size with regular shape, and typically consisted of scattered metallurgical pores and keyhole pores. This specimen exhibited excellent tensile properties and the failure mode was ductile fracture. The deformation of the defects are confined to themselves, compared to the defects located on other positions, the defects located in the plastic deformation zone prone to violent tearing expansion. For Specimen 1000, the defects were primarily irregular LOF pores distributed between the deposition layers. An increase in scanning speed resulted in a significant degradation of the tensile properties and the failure mode shifted to brittle fracture. Defects tended to aggregate between layers, and those located at the edge were prone to cracking.

## Figures and Tables

**Figure 1 materials-16-02956-f001:**
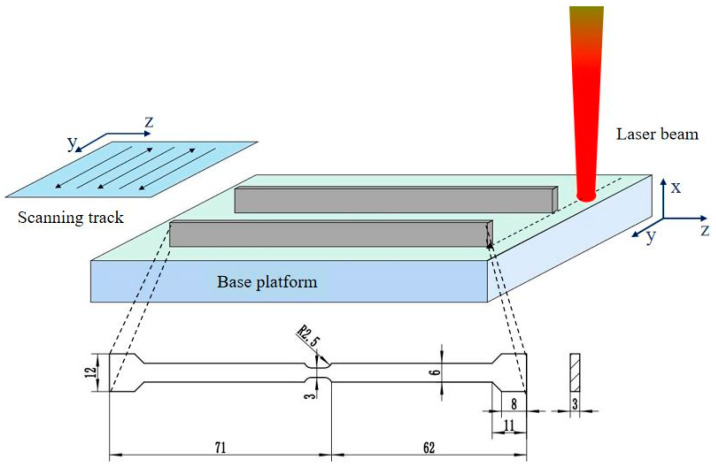
The detailed preparation process and dimensions of Invar 36 alloy specimens.

**Figure 2 materials-16-02956-f002:**
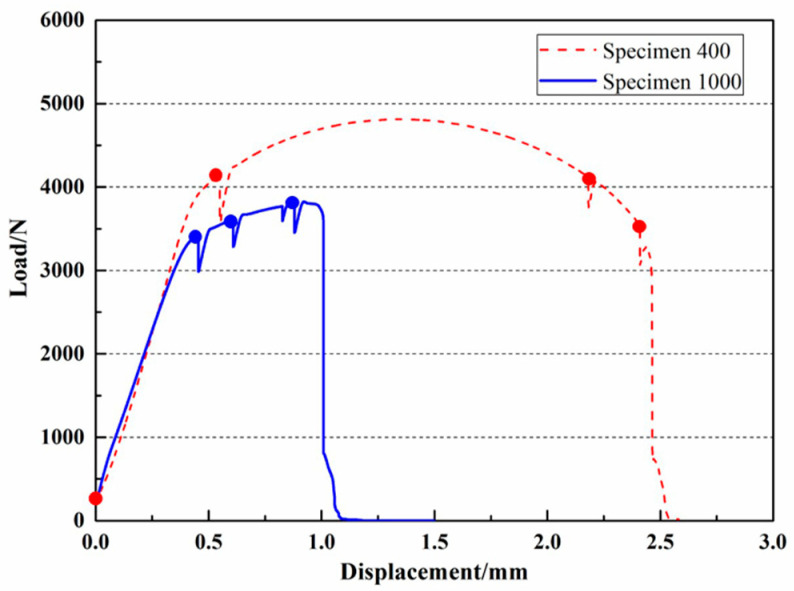
Load-displacement curves for the specimens fabricated at different scanning speeds.

**Figure 3 materials-16-02956-f003:**
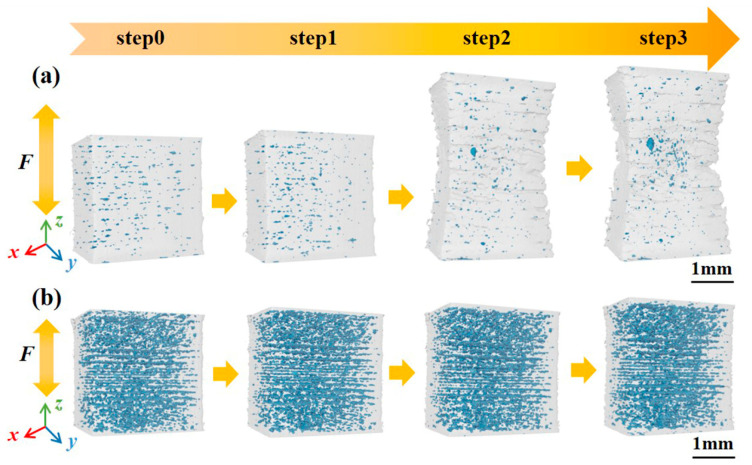
Volume renderings of VOI2 at different loading steps: (**a**) Specimen 400 and (**b**) Specimen 1000.

**Figure 4 materials-16-02956-f004:**
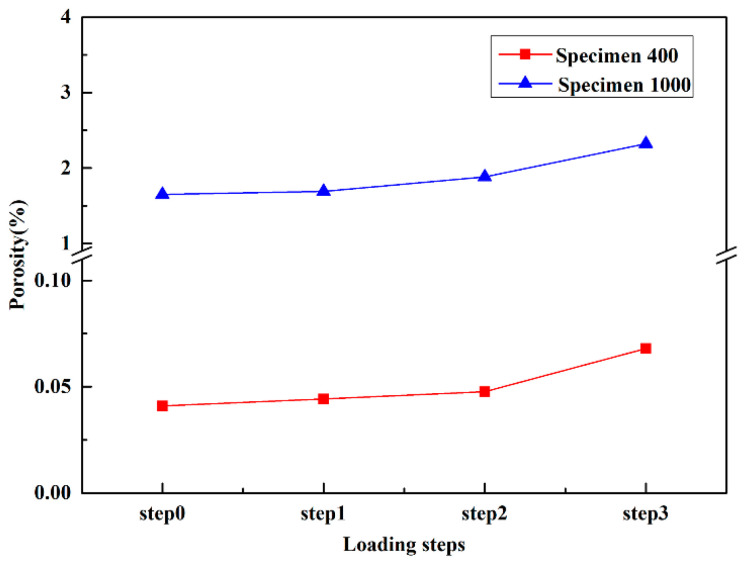
Porosity evolution of Specimen 400 and Specimen 1000 at different loading steps.

**Figure 5 materials-16-02956-f005:**
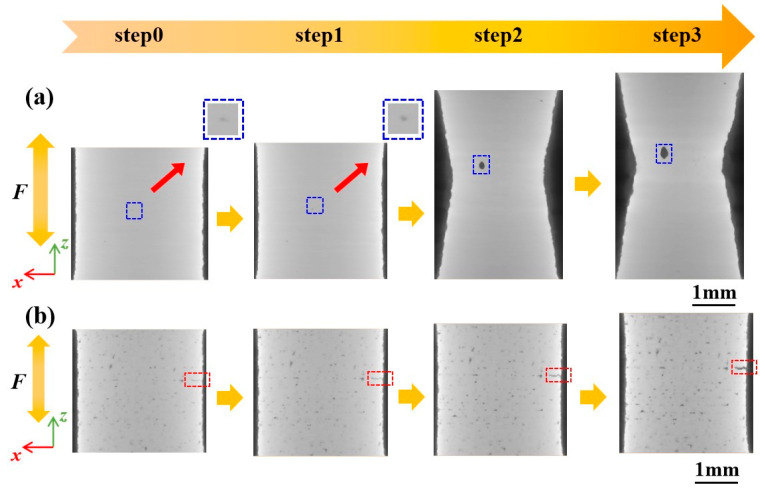
Visualization of critical defect evolution in gauge sections for (**a**) Specimen 400 and (**b**) Specimen 1000.

**Figure 6 materials-16-02956-f006:**
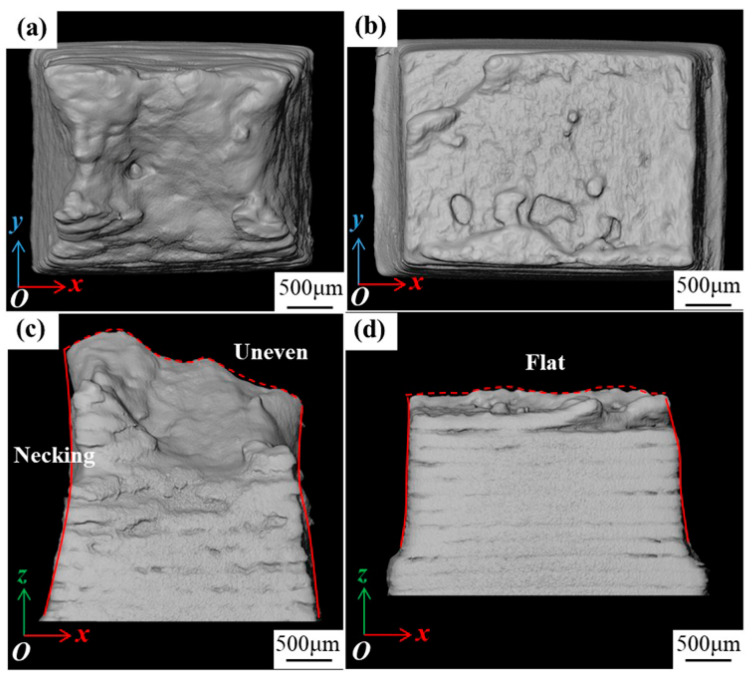
Fracture morphology of (**a**,**c**) Specimen 400 and (**b**,**d**) Specimen 1000 obtained by XCT reconstruction.

**Figure 7 materials-16-02956-f007:**
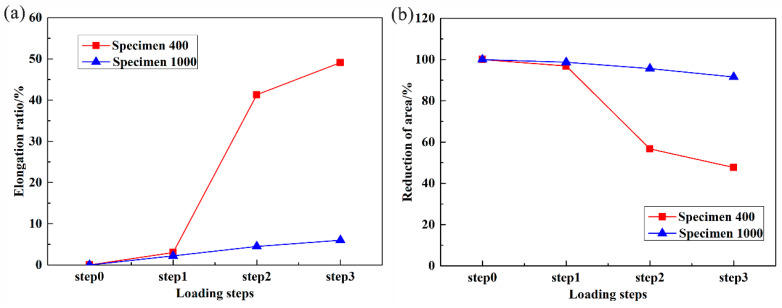
(**a**) Elongation ratios at different loading steps; (**b**) reductions of the area at different loading steps.

**Figure 8 materials-16-02956-f008:**
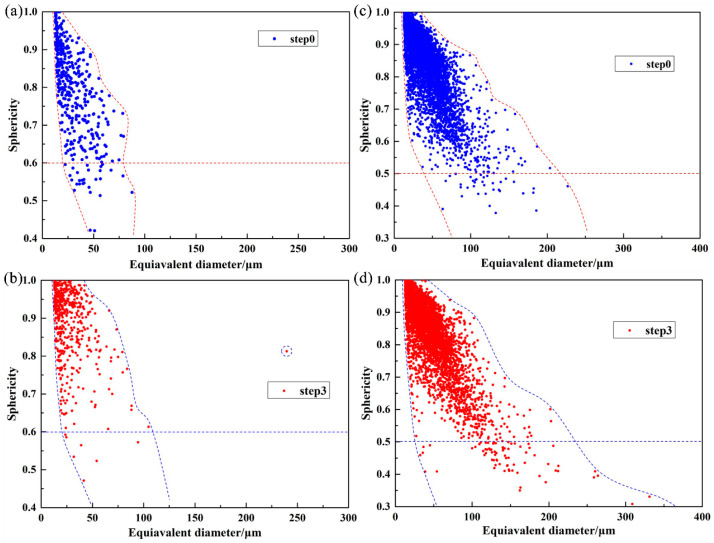
The relationships between equivalent diameter *d* and sphericity *S* for: (**a**) Specimen 400 at step0, (**b**) Specimen 400 at step3, (**c**) Specimen 1000 at step0, and (**d**) Specimen 1000 at step3.

**Figure 9 materials-16-02956-f009:**
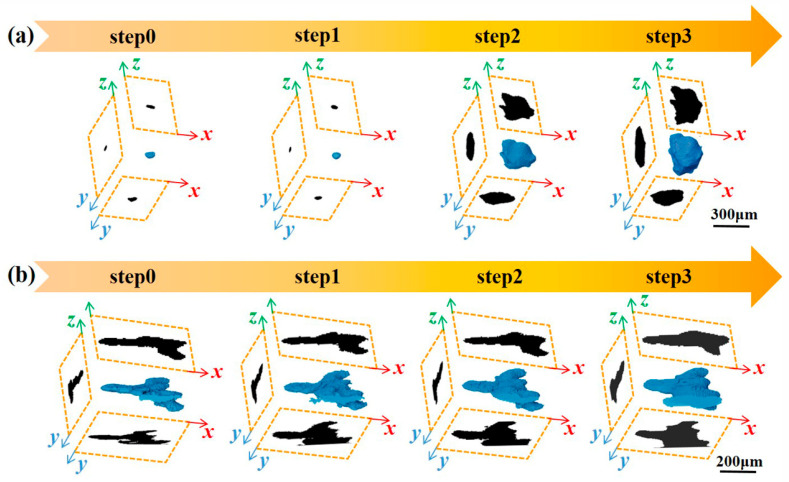
Three dimensional morphology of the critical defects of (**a**) Specimen 400 and (**b**) Specimen 1000 at different loading steps.

**Figure 10 materials-16-02956-f010:**
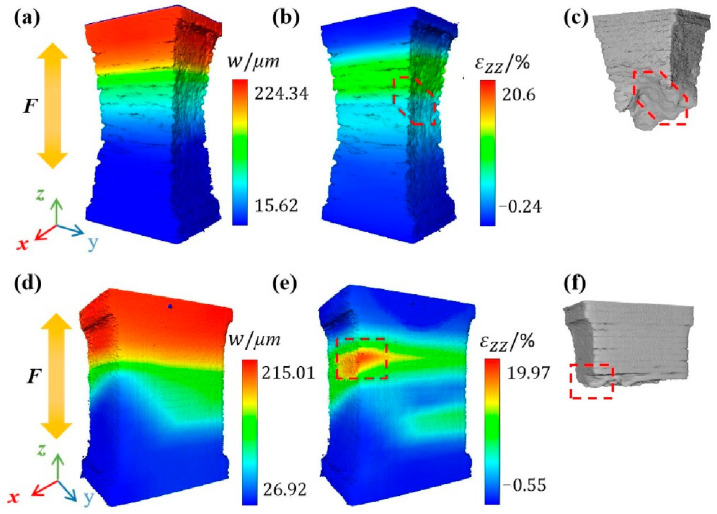
The displacement (*W*) field, uniaxial strain (ε*zz*) field and 3D rendering image of fracture of the VOI1 in (**a**–**c**) Specimen 400 and (**d**–**f**) Specimen 1000.

**Figure 11 materials-16-02956-f011:**
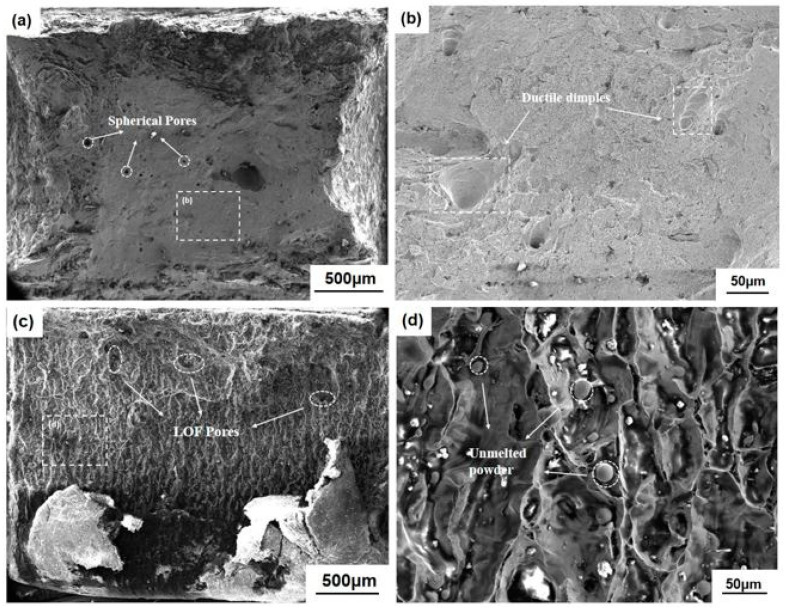
Fracture morphology of (**a**,**b**) Specimen 400 and (**c**,**d**) Specimen 1000 obtained by SEM.

## Data Availability

All the supporting and actual data are presented in the manuscript.

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
