# Peer review of "Influence of Manufacturing Defects on Mechanical Behavior of the Laser Powder Bed Fused Invar 36 Alloy: In-Situ X-ray Computed Tomography Studies"

_materials, 2023, doi:10.3390/ma16082956_

Round 1
Reviewer 2 Report
This manuscript presents a study that investigated the influence of manufacturing defects on the mechanical behavior of Invar 36 alloy produced using the LPBF technique. X-ray computed tomography (XCT) was used to examine the defects and their changes at each interruption during tensile testing, while the LPBF scanning speed was varied by 400 and 1000 mm/s. The results of defects and fracture are presented and discussed. I have some comments for the authors to consider, as follows:
Comment #1
For both cases of scanning speed (400 mm/s and 1000 mm/s), could you please calculate and show, or mention, the linear energy density (LED), which is the ratio between the power of the laser source and the scanning velocity?
Comment #2
Could you please provide a clearer explanation of the definition and the difference between VOI1 and VOI2?
Comment #3
Refer to Topic 2.2, line 102: "Firstly, the specimens were loaded to 50 N for the initial (unloaded) scan.Subsequently, tensile loads were applied to the specimens using a displacement control mode, and the tensile tests were interrupted in three times to conduct tomographic scans."
Could you explain in more detail what the criteria are for stopping a tensile test after each of the three interruptions?
Comment #4
For the caption of Figure6. "Fracture morphology of (a, b) Specimen 400 and (c, d) Specimen 1000 obtained by XCT 192 reconstruction.", please check the correctness.
Should it be revised from ".....(a, b) Specimen 400 and (c, d) Specimen 1000...." to "....(a, c) Specimen 400 and (b, d) Specimen 1000...." ?
Comment #5
Refer to Line 265: "For Specimen 1000, the critical defect is manifested as lamellar defects, which are mainly located among the deposition layers and perpendicular to the loading direction."
If available, could you please provide information about the depth and shape of the melt pool for both scanning speeds, and also discuss how they compare to the layer thickness of 30 microns?
Comment #6
Are there any disadvantages or limitations to using the low scanning speed of 400 mm/s or below, in terms of material properties, quality, or melt pool characteristics, when compared to high scanning speeds?
Reviewer 3 Report
Remarks: There are no principal essential remarks to the article. However, there are a number of comments on the presentation and design of the material:
1. At the end of the section “1. Introduction” usually contains a paragraph that reveals what research is being done in this article. It is advisable to highlight such a paragraph from the main text (from line 70).
2. In section "2.1 Materials and specimens" on line 89, the method of cutting specimens should be more clearly defined.
3. In section “5. Conclusions” in line 304, it should be more clearly stated that only samples obtained at two scanning speeds, 400 and 1000 mm/s, were examined.
4. In section “5. Conclusions" in line 312, the phrase should be formulated in such a way that it is clear that it is not defects that are prone to cracking, but samples in the zone of such defects.
